# Chondroma Arising from the Temporomandibular Joint: A Case Report

**DOI:** 10.3390/medicina59050842

**Published:** 2023-04-26

**Authors:** Masayoshi Hijiya, Masamitsu Kono, Katsuya Okuda, Shunji Tamagawa, Takuro Iyo, Tetsuya Kinoshita, Hideki Sakatani, Masanobu Hiraoka, Fumiyoshi Kojima, Shin-Ichi Murata, Muneki Hotomi

**Affiliations:** 1Department of Otorhinolaryngology-Head and Neck Surgery, Wakayama Medical University, 811-1 Kimiidera, Wakayama 641-8510, Japan; 2Otorhinolaryngology-Head and Neck Surgery, Kinan Hospital, 46-70 Shinjocho, Tanabe 646-8858, Japan; 3Department of Human Pathology, Wakayama Medical University, 811-1 Kimiidera, Wakayama 641-8510, Japan

**Keywords:** chondroma, head and neck, temporomandibular joint, parotid glands

## Abstract

Periarticular chondromas are common in the humerus and femur but rarely occur in the temporomandibular joint. We report a case of a chondroma in the anterior part of the ear. One year prior to his visit, a 53-year-old man became aware of swelling in the right cheek region which gradually increased in size. In the anterior part of the right ear, there was a palpable 25 mm tumor, elastic and hard, with poor mobility and without tenderness. A contrast-enhanced computed tomography CT showed a mass lesion with diffuse calcification or ossification in the upper pole of the parotid gland and areas of poor contrast within. A magnetic resonance imaging showed a low-signal mass lesion at the parotid gland with some high signals in both T1 and T2. Fine-needle aspiration cytology did not lead to diagnosis. Using a nerve monitoring system, the tumor was resected with normal tissue of the upper pole of the parotid gland in the same way as for a benign parotid tumor. Distinguishing between pleomorphic adenoma, including diffuse microcalcification of the parotid gland and cartilaginous tumors of the temporomandibular joint, may be sometimes difficult. In such cases, surgical resection may be a beneficial treatment option.

## 1. Introduction

Chondromas (synovial chondromatosis) are benign tumors of a mesenchymal origin that consist of mature hyaline cartilage nodules. Depending on their origin, they are classified into three clinical types. Enchondromas make up 80% of the total cases. They occur in the medulla and represent variants of condylar hyperplasia and metaphysis [1]. Juxta-articular or periosteal chondromas are located in the periphery of the joint [1,2,3]. Extra-skeletal chondromas or soft-tissue chondromas exhibit no attachment to the underlying bone or periosteum and are found most frequently in the hands and feet [4,5,6,7,8]. In general, chondromas are thought to arise from the articular synovium or from the connective tissue adjacent to the joint capsule.

The temporomandibular joint (TMJ) is a rare site of occurrence for chondromas. Conversely, the preauricular region is the most common site for parotid gland tumors, such as pleomorphic adenoma, Warthin’s tumor, or a parotid carcinoma, but not a chondroma. Furthermore, patients with a chondroma of the TMJ sometimes complain of clicking, pain of the TMJ, or difficulties in opening their mouth due to temporomandibular disorders. Patients with a chondroma of the TMJ may therefore initially be diagnosed with parotid gland tumors or inflammation of the TMJ. Herein, we report a rare case of juxta-articular chondroma that was in contact with the TMJ and partially extended to the upper pole of the parotid gland. The current case presents a difficult differential diagnosis of periauricular tumors between a benign tumor of the upper pole of the parotid gland and cartilaginous tumors derived from the TMJ and a surgical approach based on a procedure for a parotid benign tumor.

## 2. Case Presentation

A 53-year-old man presented an elastically hard, poorly mobilizable tumor (25 × 25 mm) in the right anterior auricular region. The tumor had gradually enlarged over the course of a year. There was no evidence of facial nerve palsy, clicking, pain, or mouth-opening difficulties due to temporomandibular disorders.

Computed tomography (CT) showed an oval tumor in the region of the right TMJ, extending to the upper pole of the parotid gland. The tumor was oval-shaped with an actual width of 2.5 cm and a length of 3 cm. It had well-defined borders and a diffuse hyper-absorptive zone within, which indicated diffuse calcification (Figure 1A,B). Positron emission tomography with ^18^F-fluorodeoxyglucose (^18^F-FDG-PET) showed a mild uptake of ^18^F-FDG (SUV_max_ = 3.25) (Figure 1C). There was no uptake in other sites except for physiological uptake. The tumor showed a poor contrast effect with iodine-based contrast medium. A magnetic resonance image (MRI) showed a well-defined mass in the upper pole of the right parotid gland. T1-weighted images showed a low-signal tumor (Figure 1D), while T2-weighted images showed a low-signal tumor with a macular high-signal area (Figure 1E). The contrast effect of gadolinium was poor (Figure 1F). Fine-needle aspiration cytology confirmed only neutrophils and histiocytes; there were no definitive findings.

Considering the possibility that the temporal branch of the facial nerve might have run near the tumor, we performed the same surgical procedures as those that would be used for a parotid gland tumor. Briefly, after a preauricular S-shaped skin incision, the capsule of the superficial parotid gland was exposed. Using a facial nerve monitoring system, NIM-NEURO™3.0 (Medtronic Japan Co., Ltd., Tokyo, Japan), the main trunk of the facial nerve was confirmed near the stylomastoid foramen. The temporal branch of the facial nerve was exposed in a peripheral direction. The tumor was located at the upper pole of the parotid gland and was in contact with the TMJ capsule and the temporal branch of the facial nerve. Meanwhile, direct contact with the external auditory canal was unclear, and there was no obvious adhesion to the surrounding parotid tissue or facial nerve (Figure 2A). The tumor was resected with the upper pole of the superficial parotid gland, and the temporal branch of the facial nerve was preserved with a normal response through the use of the nerve monitoring system (Figure 2B). The surface of the tumor was well-demarcated and completely encapsulated by a smooth capsule, and half of it was buried within the upper pole of the parotid gland (Figure 2C). The interior of the tumor was white and contained numerous calcifications (Figure 2D). To rule out malignant diseases, an intraoperative consultation with a frozen section was performed. Neoplastic tissue with a chondromyxoid matrix and a calcified deposition was observed without distinct malignant findings.

Microscopically, a well-circumscribed lobulated tumor was found within the parotid gland. The cellularity in the lesion was low, and isolated basophilic cells with indistinct cytoplasmic borders were scattered in the eosinophilic to basophilic chondroid matrix. Calcified materials were diffusely deposited in the tumor. Lymphocytic infiltration with foreign-body giant cells and a few Touton giant cells were distributed mainly along the interlobular septa and in the periphery of the tumor (Figure 3A). Direct Fast Scarlet staining (DFS) did not demonstrate amyloid deposition (Figure 3B). Von Kossa staining identified calcified materials (Figure 3C). Immunohistochemically, the tumor cells were positive for CD68, focally positive for CD163, weakly positive for S100 protein, and negative for Cytokeration (AE1/AE3), p63, and α-smooth muscle action (SMA) (Figure 3D). The final pathological diagnosis was chondroma with prominent calcification.

After surgery, the patient developed mild palsy of the temporal branch of the facial nerve and was treated with oral prednisolone with a dose tapered from 30 mg for one week. The cause of the paralysis was thought to be nerve extension due to the detachment procedure during the surgery. This facial palsy was completely healed two weeks after the surgery. The patient was followed with an ultrasound examination every three months for the first year and then every six months for two years. A plain MRI examination at six months and two years after the surgery confirmed no evidence of recurrence. At the time of the final checkup, both oral and written consent were obtained from the patient, and the Wakayama Medical University Institutional Review Board approved publication of this case report (protocol code 3814, approved on 17 March 2023).

## 3. Discussion

Chondromas usually occur proximal to articular capsules and tendons and rarely occur outside of the skeleton or in soft tissues. These tumors are thus rarely found in the TMJ and preauricular region [8,9]. Chondromas occurring in the TMJ account for only 3% of all cases of chondroma. The most common age and sex of this disease are thought to be patients in their 40s and female, respectively. Song et al. recently performed a systematic review of chondroma of the TMJ [10]. Among 188 cases, 70.7% were female, and 34% showed bony changes including calcification. Major clinical symptoms were pain (65.4%), limited mouth opening (53.7%), swelling (27.7%), crepitus (19.7%), and clicking of the TMJ (10.1%). A diagnosis of chondroma of the TMJ can be difficult due to its rarity, painless nature, slowly progressive growth, and inconspicuous radiologic appearance. According to this review, our reported case of chondroma, which probably developed from the TMJ, is markedly unusual.

Pleomorphic adenomas, the most common type of benign parotid tumor, may contain chondrogenic differentiation. Owing to the difficulty in distinguishing from cartilage by imaging, strict morphologic criteria are necessary to rule out benign adenomas. Chondromatous degeneration via the accumulation of mucoid material around the myoepithelial cells in pleomorphic adenomas of parotid glands may resemble a chondroma. Coelho et al. described the characteristics of the imaging of pleomorphic adenoma with calcification as follows: “(i) a smoothly marginated mass that is hypoechoic on ultrasound, (ii) a higher degree of attenuation in the tumor than in the surrounding parenchyma on CT, and (iii) low T1-weighted and high T2-weighted signal intensity on magnetic resonance imaging (MRI)” [11]. If the tumor extends into the parotid gland, distinguishing chondroma from pleomorphic adenoma would become particularly difficult. In the present case, part of the tumor was in contact with the temporomandibular joint capsule but not with the external auditory canal cartilage. The tumor was therefore thought to have originated from the cartilaginous tissues of the TMJ and then gradually extended into the upper pole of the parotid gland. In addition, because the TMJ anatomically consists of the temporomandibular fossa and the mandibular head, with the articular disc composed of cartilage, chondromas are sometimes caused by damage to the articular disc due to localized external pressure. The complication of TMJ arthropathy is also an important etiological factor, although it was not clearly observed in the present case.

Chondrosarcoma is also a relatively rare type of tumor, but it should be ruled out for the differential diagnosis of malignant diseases around the TMJ. In general, chondrosarcoma pathologically shows increased cellularity with atypical findings, such as binucleated chondrocytes. In the current case, some atypical cells with unclear cell boundaries were observed. Some cases of chondrosarcoma were thought to arise via the malignant transformation of chondroma [12]. Recently, some genes were found to be important for the malignant transformation of chondroma by activating mesenchymal transition and the VEGFA-VEGF2R signaling pathway [13]. MRI findings in the malignant transformation of chondroma, bone edema, periosteal reaction, and soft tissue edema are considered significant [14]. Fine-needle aspiration cytology is usually useful for estimating the malignancy of head and neck tumors; however, it is not applicable to those originating from cartilage if not enough cells are available. ^18^F-FDG-PET is reported to be a useful examination for evaluating the potential or grade of the malignancy of chondroid tumors [15]. SUVmax is correlated with histologic grade in chondroid tumors; a very low SUVmax value (1.6 ± 0.7) supports a diagnosis of benign tumor, while an elevated SUVmax (4.4 ± 2.5) value is suggestive of a higher grade chondrosarcoma.

The pathogenesis of chondromas is thought to be osteochondromal exostosis or the proliferation of germ cells with chondrogenic potential due to abnormalities in the direction of osteochondral growth. Calcification occurs as a result of development or regression of the lesion over a long period of time [16,17]. Histologically, chondromas are characterized by a lobulated mass of partially subdivided hyaline cartilage nodules surrounded by a fibrous stroma, often with calcified foci and ossification. In general, the synovium is diffusely studded with many nodules that are polypoid or pedunculated with a delicate stalk and variable in size. Chondromas in the TMJ have been described as follows; “Synovial chondromatosis shows the presence of nodules of mature cartilage of varying cellularity within the synovium and lying loosely in the joint space. The cartilage may appear atypical, with hypercellularity, hyperchromasia, binucleated chondrocytes, and increased mitoses. Calcification and ossification may be present” [18]. Milgram proposed classification into three stages based on histological findings, as follows; Stage I: chondrogenesis occurs in synovial mesenchymal stem cells, with no formation of loose bodies; Stage II: a transitional lesion with active chondrogenesis in the synovium and cartilaginous nodules enlarge or are released into the articular cavity as loose bodies; Stage III: chondrogenesis in the synovial membrane disappears and loose bodies can be observed [10,19].

Histologically, most extraskeletal chondromas have a mature hyaline cartilaginous appearance, and approximately one-third exhibit focal or diffuse calcification. In our patient, diffuse prominent calcification and fibrohistiocytic reaction obscured the chondroid appearance (hyaline cartilaginous matrix and lacuna of chondrocyte). These findings made this case mimic tumoral calcinosis and made the diagnosis difficult. It is important to detect cartilage with careful observation for the differentiation between tumoral calcinosis and an extraskeletal chondroma. Immunohistochemically, chondrocytes are positive for S100, while epithelial and myoepithelial cells, which are not typically present in chondromas, are negative for S100. In the current case, there were cells that were weakly positive for S100, but there were no epithelial or basal cells that were positive for CK and p63. In addition, the von Kossa stain was positive; therefore, the current patient was diagnosed as having a chondroma with prominent calcification. Moreover, this case was considered a juxta-articular or periosteal chondroma in close proximity to the TMJ because it was located posterolaterally to the mandibular condyle and had no contact with the articular surface.

Fine-needle aspiration cytology is a useful and less-invasive examination for the preoperative diagnosis for head and neck neoplasms, even for cartilaginous tumors. The *WHO Classification of Tumours (5th Edition), Soft Tissue and Bone Tumours*, 2020, describes findings on the cytology of cartilagenous tumors [20]. If a tumor contains both mucinous and liquid regions, it is possible to prepare specimens for cytological evaluation from both regions. The general cytological appearance of a chondroma comprises tumor cells with small pyknotic nuclei with a hyaline-like cartilage matrix. In our patient, however, the tumor was stony hard and consisted only of solid components and calcifications; thus, a sufficient number of cells could not be obtained by-fine needle aspiration for diagnosis.

As a policy for surgery, we decided to treat using the same strategy for benign tumors of the parotid gland because of the relatively high recurrence rate of chondromas, including other sites in previous reports. Additionally, some pleomorphic adenomas are accompanied by calcification within the tumors [11,21], which makes preoperative diagnosis difficult. In this case, although there was no parotid tissue on the upper side of the tumor, the exposed capsule was preserved. On the other hand, the caudal side was located in the upper pole of the parotid gland. We were able to completely remove the lesion by resecting it, including the normal tissue of the upper pole of the parotid gland. Preservation of the temporal branch of the facial nerve is important when performing surgery for tumors around the temporomandibular joint due to the anatomical positional relationship. In our patient, the nerve was extensively in contact with the tumor, requiring careful detachment. The nerve monitoring system worked effectively for saving this thin nerve. Local surgical resection is our recommended treatment; however, as chondromas are very rare, an accumulation of cases is necessary to define the most appropriate treatment.

## 4. Conclusions

This case demonstrates the difficulty in differential diagnosis between a pleomorphic adenoma with chondrogenic differentiation of the upper pole of the parotid gland and a chondroid tumor of the TMJ. Differentiation via imaging findings is sometimes difficult, and diagnosis requires immunological findings with pathology if it is difficult to obtain enough cells by fine-needle aspiration.

Surgical resection is recommended in such cases because of the possibility of malignant transformation and the relatively high recurrence rate, including occurrence in other sites, even in the case of benign chondroid species. Intra-operative nerve monitoring is helpful to preserve temporal branch of the facial nerve if the tumor is located inside the upper pole of parotid gland.

## Figures and Tables

**Figure 1 medicina-59-00842-f001:**
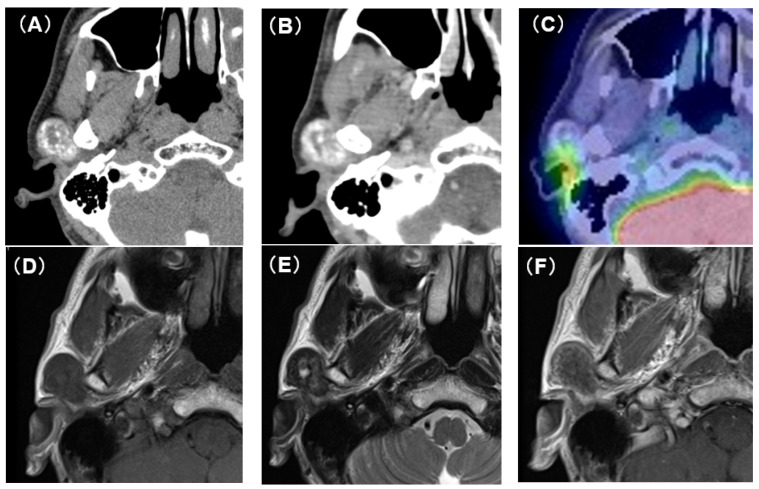
CT, PET-CT, and MRI findings. (**A**) Plain CT horizontal section. The inside of the tumor is diffusely accompanied by a high-density area. (**B**) Contrast-enhanced CT horizontal section. The contrast enhancement effect of the tumor is poor. (**C**) PET-CT image findings. The tumor showed mild uptake of 18F-FDG (SUV_max_ = 3.25). (**D**) Plain MRI T1 image findings. The tumor is visualized as an area of uniform low-intensity. (**E**) Plain MRI T2 image findings. Low-intensity mass lesion with faint patchy hyperintensity inside. (**F**) Contrast-enhanced MRI T1 image findings. The tumor enhancement with gadolinium was poor.

**Figure 2 medicina-59-00842-f002:**
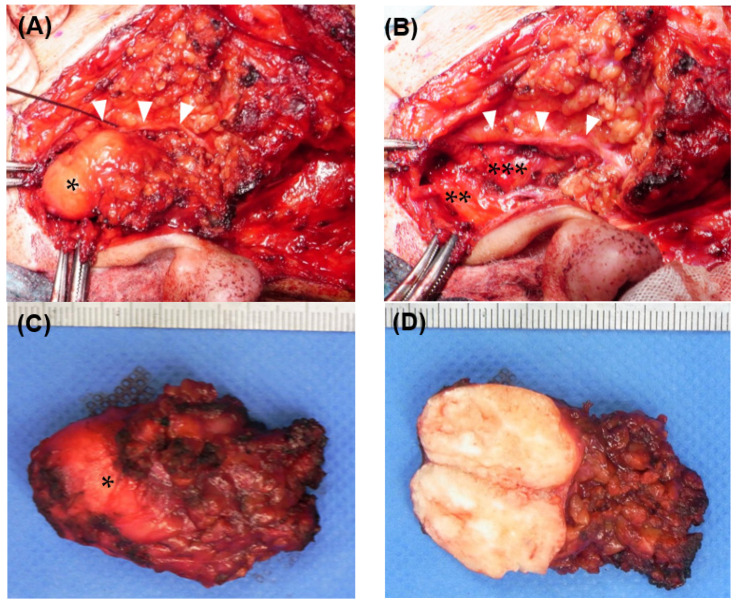
Intraoperative findings (**A**) Tumor exposure. The tumor (*) was located at the upper pole of the parotid gland. The temporal branch of the facial nerve (white arrowheads) runs widely along the anterior side of the tumor. (**B**) After tumor resection. The tumor was in contact with the external auditory canal cartilage (**) and mandibular condyle (***), but there was no obvious adhesion. The temporal branch of the facial nerve (white arrowheads) was preserved. (**C**) The resected tumor. It was 25 mm in length and 15 mm in width. Although half of the cranial side was exposed, the tumor was encapsulated with a rigid capsule (*). (**D**) The tumor section. There were numerous coarse calcifications inside the tumor.

**Figure 3 medicina-59-00842-f003:**
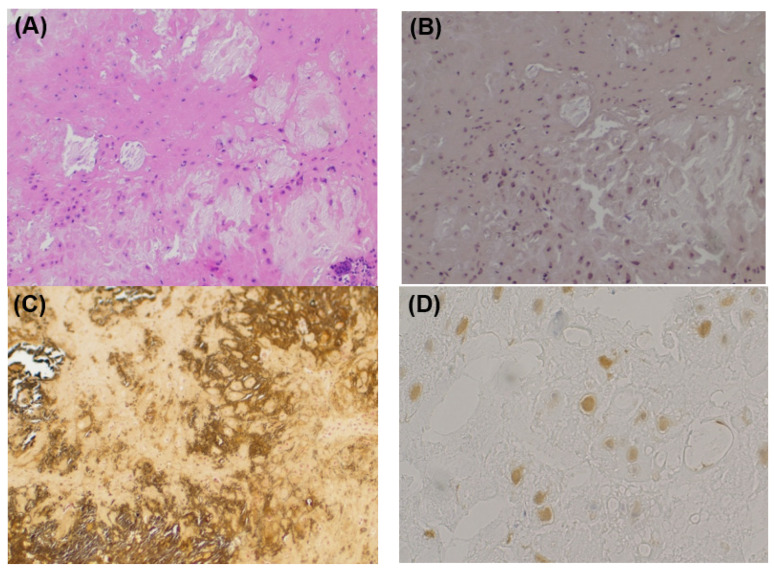
Histopathological findings. (**A**) HE staining. In the background of eosinophilic matrix with calcification, cells without atypia with unclear cell boundaries are scattered. Multinucleated giant cell and lymphocyte infiltration are observed. (**B**) DFS staining. The DFS staining is negative. There is no evidence of amyloid deposition. (**C**) Kossa staining. Calcified materials are identified in black. (**D**) S100 staining. Some tumor cells are weakly positive for S100 protein.

## Data Availability

The data presented in the present study are available on request from the corresponding author.

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
