# Peer review of "Chondroma Arising from the Temporomandibular Joint: A Case Report"

_medicina, 2023, doi:10.3390/medicina59050842_

Round 1
Reviewer 1 Report
The authors provide the report of an interesting case of a patient presenting with a mass located next to the temporomandibular joint, which presents a difficult differential diagnosis between pleomorphic adenoma of the parotid gland and cartilagineous tumor. Anamnesis, diagnostic examinations, surgical treatment and pathological findings are clearly presented and illustrated with appropriate figures. The discussion clearly explains the difficulties of defining the nature of the mass and points out the recommended treatment.
There are a few minor remarks:
- Line 35 and 119: a (**) mark appears, but it is not clear if it refers to the ** marks in figure 2 or if it was a placeholder for a reference number;
- Line 63: "This is a figure. Schemes follow the same formatting." should be deleted;
- Line 106: "chondrom" should be "chondroma".
Author Response
Dear Reviewer 1,
First of all, we would like to thank the reviewer for the careful review of our manuscript. We are very sorry for lacking the information about ethical review. The date of approval, the name of the review board, and the protocol code were included in the revised manuscript. We also significantly increased the manuscript as our manuscript did not meet the submission requirement of 2500 words or more. Please see the answers below for reviewers' comments.
- Line 35 and 119: a (**) mark appears, but it is not clear if it refers to the ** marks in figure 2 or if it was a placeholder for a reference number;
We apologize this. ** marks in the manuscript are deleted.
- Line 63: "This is a figure. Schemes follow the same formatting." should be deleted;
The reviewer is correct. This description is deleted.
- Line 106: "chondrom" should be "chondroma".
The reviewer is correct. We corrected this misspelling.
Reviewer 2 Report
The manuscript by Hijiya M et al reports on a case of a juxta-articular chondroma in a 53-year-old male patient. The manuscript is well-written, and the case is described satisfactorily.
Some minor changes should be addressed:
I suggest adding "A case report:" to the end of the manuscript title to clearly indicate the type of article.
The following sentence can be rephrased for clarity: "In addition, because anatomically the temporomandibular joint consists of the temporomandibular fossa and the mandibular head with the articular disc composed of cartilage in between chondromas are sometimes caused by damage to the articular disc due to localized external pressure"
Also "Immunohistochemically, chondrocytes possess S100, although epithelial and myoepithelial cells are negative for S100" should be rephrased. The meaning of the sentence is different if you say “Immunohistochemically, chondrocytes are positive for S100, while epithelial and myoepithelial cells, which are not typically present in chondromas, are negative for S100.” which is what the authors, I guess, wanted to express.
If found two "(**)" in the manuscript without any corresponding citation, it may indicate that a citation was intended but was not included in the text.
Authors describe the tumor histology as “poorly heterotypic cells were accompanied by multinucleated giant cells and foam cell infiltration” and “ atypical cells with unclear cell boundaries proliferate sparsely”. Please, provide further discussion to address these findings that, in my opinion, are more likely to be found in chondrosarcoma.
Author Response
Dear Reviewer 2,
First of all, we would like to thank the reviewer for the careful review of our manuscript. We are very sorry for lacking the information about ethical review. The date of approval, the name of the review board, and the protocol code were included in the revised manuscript. We also significantly increased the manuscript as our manuscript did not meet the submission requirement of 2500 words or more. Please see the answers below for reviewers' comments. In the revised manuscript, corrections/modifications made in response to the reviewer’s comments are indicated in red.
I suggest adding "A case report:" to the end of the manuscript title to clearly indicate the type of article.
We agree the reviewer’s suggestion. The title was changed as “Chondroma arising from the temporomandibular joint: A case report”.
The following sentence can be rephrased for clarity: "In addition, because anatomically the temporomandibular joint consists of the temporomandibular fossa and the mandibular head with the articular disc composed of cartilage in between chondromas are sometimes caused by damage to the articular disc due to localized external pressure"
The reviewer is correct. There is a grammatical problem in this description. We corrected as below (Lines 184-187);
“In addition, because anatomically the TMJ consists of the temporomandibular fossa and the mandibular head with the articular disc being composed of cartilage, chondromas are sometimes caused by damage to the articular disc due to localized external pressure.”
Also "Immunohistochemically, chondrocytes possess S100, although epithelial and myoepithelial cells are negative for S100" should be rephrased. The meaning of the sentence is different if you say “Immunohistochemically, chondrocytes are positive for S100, while epithelial and myoepithelial cells, which are not typically present in chondromas, are negative for S100.” which is what the authors, I guess, wanted to express.
We appreciate the reviewer’s suggestion. This part was replaced as the reviewer’s description. (Lines 227-229)
If found two "(**)" in the manuscript without any corresponding citation, it may indicate that a citation was intended but was not included in the text.
We apologize this. ** marks in the manuscript are deleted.
Authors describe the tumor histology as “poorly heterotypic cells were accompanied by multinucleated giant cells and foam cell infiltration” and “ atypical cells with unclear cell boundaries proliferate sparsely”. Please, provide further discussion to address these findings that, in my opinion, are more likely to be found in chondrosarcoma.
The reviewer is correct. This description was wrong and replaced as below. (Lines 151-153)
“In the background of eosinophilic matrix with calcification, cells without atypia with unclear cell boundaries are scattered. Multinucleated giant cell and lymphocyte infiltration is observed.”